# Biodiversity and Spatial-Temporal Dynamics of *Margalefidinium* Species in Jiaozhou Bay, China

**DOI:** 10.3390/ijerph182111637

**Published:** 2021-11-06

**Authors:** Shuya Liu, Mengjia Zhang, Yongfang Zhao, Nansheng Chen

**Affiliations:** 1CAS Key Laboratory of Marine Ecology and Environmental Sciences, Institute of Oceanology, Chinese Academy of Sciences, Qingdao 266071, China; liushuya@qdio.ac.cn (S.L.); zhangmengjia@qdio.ac.cn (M.Z.); 2Functional Laboratory of Marine Ecology and Environmental Science, Qingdao National Laboratory for Marine Science and Technology, Qingdao 266237, China; 3Center for Ocean Mega-Science, Chinese Academy of Sciences, Qingdao 266071, China; 4College of Marine Science, University of Chinese Academy of Sciences, Beijing 100039, China; yfzhao@qdio.ac.cn; 5Jiaozhou Bay National Marine Ecosystem Research Station, Institute of Oceanology, Chinese Academy of Sciences, Qingdao 266071, China; 6Department of Molecular Biology and Biochemistry, Simon Fraser University, Burnaby, BC V5A 1S6, Canada

**Keywords:** *Margalefidinium*, metabarcoding analysis, Jiaozhou Bay, harmful algal bloom, spatial-temporal dynamics

## Abstract

Many *Margalefidinium* species are cosmopolitan harmful algal bloom (HAB) species that have caused huge economic and ecological damage. Despite extensive research on *Margalefidinium* species, the biodiversity and spatial-temporal dynamics of these species remain obscure. Jiaozhou Bay is an ideal area for HAB research, being one of the earliest marine survey areas in China. In this study, we carried out the first metabarcoding study on the temporal and spatial dynamics of *Margalefidinium* species using the 18S rDNA V4 region as the molecular marker and samples collected monthly at 12 sampling sites in Jiaozhou Bay in 2019. Two harmful *Margalefidinium* species (*M. polykrikoides* and *M. fulvescens*) were identified with potentially high genetic diversity (although we cannot rule out the possibility of intra-genome sequence variations). Both *M. polykrikoides* and *M. fulvescens* demonstrated strong temporal preference with a sharp peak of abundance in early autumn (September), but without showing strong location preference in Jiaozhou Bay. Our results revealed that temperature might be the main driver for their temporal dynamics. Knowledge of biodiversity and spatial-temporal dynamics of the *Margalefidinium* species may shed light on the understanding of mechanisms underlying strongly biased occurrences of *Margalefidinium* blooms recorded globally.

## 1. Introduction

The genus *Margalefidinium*, which contains five species, namely, *M. polykrikoides*, *M. fulvescens*, *M. catenatum*, *M. citron*, and *M. flavum*, was recently transferred from the genus *Cochlodinium* on the basis of morphological and phylogenetic analyses [1]. It belongs to the family Gymnodiniaceae, order Gymnodiniales, class Dinophyceae, and phylum Miozoa [2]. Among these five described *Marglefidinium* species, two species, *M. polykrikoides* and *M. fulvescens*, have drawn particularly wide attention and have received extensive research primarily for their ability to form harmful algal blooms (HABs) [1,3]. Each of the five species has an eyespot in the episome that may be important for photoreception. Both of them are large (>40 μm) and unarmored, commonly form chains, and have shallow sulcus without harboring longitudinal flagellum [4,5]. They differ in chloroplast shape, eyespot, and the position of sulcus, which are discernible under light microscopy, and in morphology of apical groove, which is discernible only under scanning electron microscope (SEM) [6,7]. However, these two HAB species cannot be easily distinguished morphologically under light microscope (LM) since they share similar morphological features with few differences.

Both *M. polykrikoides* and *M. fulvescens* are widespread, with populations being documented in tropical and warm-temperate waters around the world, and blooms being expanding in scope, respectively [8]. The first blooms of *M. polykrikoides* were detected in Southeast Asia and the east coast of North America before 1990 [6,9,10]. Since then, *M. polykrikoides* blooms have expanded to the East China Sea, the Philippines, Malaysia, Australia, the west coast of North America, Costa Rica, Latin America, Europe, and the Arabian Gulf region [7,8,11]. *M. fulvescens* has been reported to form massive blooms on the coast of Vancouver Island, Canada, and along the California coast, USA [12,13]. HABs caused by *M. polykrikoides* and *M. fulvescens* were not only strongly ichthyotoxic, but also capable of killing many other marine organisms [7]. In particular, *M. polykrikoides* had caused destructive blooms on a global scale, which were responsible for mass mortalities of wild and farmed fish, with catastrophic impact on local fishery operations, tourism, and economies [7,11]. Similarly, *M. fulvescens* blooms have caused mass mortalities of farmed salmon and mussels [12,13]. Although the mechanisms of toxicity of *M. polykrikoides* to fish have yet to been fully validated, they may involve many substances, such as reactive oxygen species (ROSs), hemolytic and neurotoxic-like substances, and extracellular mucoid polysaccharide substances [8,14,15]. Further studies showed that sonicated cultures and cell-free culture media of *M. polykrikoides* exhibited significantly reduced toxicity, suggesting that toxins were unstable under extreme physical conditions [16]. *M. fulvescens* is difficult to culture, and there are relatively few studies on its harmful effects; some studies suggested that *M. fulvescens* possesses ichthyotoxic properties similar to those of *M. polykrikoides* [5,17].

Characterization of *Margalefidinium* species has been carried out largely using microscopical methods. The life cycle of *M. polykrikoides* was detected in monoclonal isolation and morphological observation, indicating two different stages (an armored and an unarmored vegetative stage) [18]. Resting cysts have been considered a fundamental attribute of dinoflagellate life cycles [19], that in *M. polykrikoides* is also a research “hot” topic. The ability of *M. polykrikoides* to produce resting cysts in culture was first confirmed in 2012, and possible ecological implication was then discussed, such as the contribution to bloom expansion and annual recurrence [20]. *M. polykrikoides* is also able to form hyaline cysts, which are immotile chain-forming cells surrounded by a transparent membrane [21]. However, there is little report about resting cysts in *M. fulvescens* in previous studies.

Among species of the genus *Margalefidinium*, *M. fulvescens* and *M. polykrikoides* may often coexist in some coastal waters [22,23]. *M. fulvescens* and *M. polykrikoides* are not easily distinguished morphologically under LM due to their similar morphological characteristics, especially after preservation [5,7]. In addition, the microscopic quantification of *Margalefidinium* species at lower abundance is highly challenging, particularly in the case of post-preservation identification, necessitating alternative approaches such as molecular tools [7]. In recent years, molecular techniques used to identify genetic diversity and provide new insight into the physiology and ecology of phytoplankton. Through monoclonal isolation and partial regions of large subunit ribosomal RNA gene (LSU rDNA) sequences, researchers divided *M. polykrikoides* strains into four clades (i.e., ribotypes): East Asian, Philippines, American/Malaysian, and Mediterranean type [8,9,17]. With the advancement of next-generation sequencing technologies, metabarcoding analysis has been demonstrated to have great potential to strengthen biological monitoring phytoplankton communities in the seawaters. Hattenrath-Lehmann and colleagues [24] conducted metabarcoding analyses using partial 16S rRNA (V4 region) and 18S rRNA (V7-8 regions) during *M. polykrikoides* bloom in USA, and found that the lower diversity of plankton communities of blooms might be due to allelopathically excluded by *M. polykrikoides*. Cui and colleagues studied microbial module regulations on the *M. polykrikoides* blooms using 16S rRNA (V3–V5 region for Archaea; V3–V4 for Bacteria) and 18S rRNA (V8–9 regions) as molecular markers [25].

*Margalefidinium* species only recently emerged as the dominant HAB species after persisting for years at low abundance of the total phytoplankton communities. For example, the first outbreak of *M. polykrikoides* in Korea was in 1981, and since then, *M. polykrikoides* blooms occurred regularly and expanded to most of the country’s coastal areas [9,26]. Similar phenomena were also found in the coastal waters of California, USA [12], and the Mediterranean Sea [10]. Thus, it was urgent to track these species distributions before their blooms outbreak, especially for the seawaters without previously recorded *Margalefidinium* blooms. Despite extensive research on *Margalefidinium* species, the biodiversity and spatial-temporal dynamics of these species remain obscure. Knowledge of biodiversity and spatial-temporal dynamics may shed light on the understanding of abrupt occurrences and continuity of *Margalefidinium* blooms in ocean regions globally.

Jiaozhou Bay, which is a semi-enclosed bay with an area of approximately 367 km^2^, located on the western coast of the Yellow Sea [27], is an excellent ocean region for carrying out research on the biodiversity and spatial-temporal dynamics of *Margalefidinium* species. *M. polykrikoides* and *M. fulvescens* have been described in Jiaozhou Bay [5,28]. On the basis of analysis of large subunit (LSU) sequences, *M. polykrikoides* in Jiaozhou Bay belongs to the East Asian ribotype [28], and *M. polykrikoides* of Jiaozhou Bay strains (MPJZB-C3 and MPJZB-D6) showed lower toxicity than that of USA strain (CP1) and Malaysia strain (MPCoKK23) [16]. Hu and colleagues [5] confirmed the presence of *M. fulvescens* in Jiaozhou Bay through morphological and phylogenetic characterization, and found that *M. fulvescens* dominated the dinoflagellate community of Jiaozhou Bay in the early autumn of 2015. Recently, Lin and colleagues [29] assessed the intra-populational and intra-genome genetic diversity of *M. fulvescens* and inferred high variability among LSU copies. Although no severe HABs caused by *Margalefidinium* species in Jiaozhou Bay has been reported [30], the potential of *Margalefidinium* bloom outbreaks is high. Thus, it is important to detect these *Margalefidinium* species in Jiaozhou Bay.

In this study, we conducted metabarcoding analysis using 18S rDNA V4 regions for samples collected monthly at 12 sites in Jiaozhou Bay in 2019, targeting *Margalefidinium* species. We confirmed the presence of *M. fulvescens* and *M. polykrikoides*, and the two species showed high genetic diversity (may be copy diversity in the individual cells). We also compared the spatial-temporal dynamics of *Margalefidinium* species, as well as the environmental factors.

## 2. Materials and Methods

### 2.1. Sampling Collection

We conducted 12 monthly expedition voyages in 2019 in Jiaozhou Bay, sampling at 12 sites during each expedition (Figure 1). At each site, 1 L of seawater was collected at different sampling depths. The number of samples collected at each sampling site depended on the water depth of the corresponding sampling site. Water filtration and storage were the same as described previously [31]. Briefly, each seawater sample was filtered using a 200 μm mesh (Hebei Anping Wire Mesh Co., Ltd., Hengshui, China) to remove large phytoplankton and debris, followed by filtration using 10 μm polycarbonate membranes (Millipore, Burlington, MA, USA) and 0.2 μm polycarbonate membranes, sequentially. The membranes with samples were stored in liquid nitrogen until they are used for DNA exactions. A total of 468 filtered membrane samples were obtained in this study.

Environmental data (temperature, salinity, SiO_3_^2−^, PO_4_^3−^, NO_2_^−^, NO_3_^−^, NH_4_^+^, and chlorophyll concentrations) were provided by Jiaozhou Bay National Marine Ecosystem Research Station, Institute of Oceanology, Chinese Academy of Sciences.

### 2.2. DNA Extraction, PCR Amplification, and Sequencing

For each membrane with filtrated samples, 500 μL CSPL buffer (Omega, Norwalk, CT, USA) was quickly added after they were taken out of liquid nitrogen. It was then cut into pieces using sterilized scissors for approximately 50 times, followed by crushing in a cell crushing apparatus (MP Biomedicals, Santa Ana, USA) for 5s at a speed of 4 m/s. DNA was subsequently extracted using the HP Plant DNA Kit (Omega, Norwalk, CT, USA), according to the manufacturer’s instructions.

The 18S rDNA v4 regions were amplified using the V4F forward primer, CCAGCA(G/C)C(C/T)GCGGTAATTCC, and the V4R reverse primer, ACTTTCGTTCTTGAT(C/T)(A/G)A [32], with a unique barcode to distinguish different samples. Each PCR amplification was performed in a final volume of 50 μL, including 50 ng template DNA (n μL), 25 μL 2X Taq PCR Master Mix (Tiangen Biotech, Beijing, China), 1 μL forward primer, and 1 μL reverse primer. The remaining volume (23-n μL) was supplemented with ddH_2_O to reach a total of 50 μL. PCR cycling conditions included an initial denaturation at 94 °C for 4 min, followed by 32 cycles of denaturation at 94 °C for 1 min, annealing at 57 °C for 1 min and 50 s, elongation at 72 °C for 2 min, and a final extension at 72 °C for 10 min. To check the product length, PCR products were monitored on 1% agarose gels, followed by purification with the Qiagen Gel Extraction Kit (Qiagen, Germany). Sequencing libraries were generated using the TruSeq^®^ DNA PCR-Free Sample Preparation Kit (Illumina, Inc., San Diego, CA, USA). Library quality was assessed on a Qubit@ 2.0 Fluorometer (Thermo Scientific, Waltham, MA, USA) and the Agilent Bioanalyzer 2100 system (Agilent Technologies, Santa Clara, CA, USA). Sequencing was conducted on the Illumina NovaSeq platform (Novogene, Beijing, China) with 250 bp paired-end reads. Raw data have been deposited to NCBI under the BioProject number PRJNA577777 and PRJNA733859.

### 2.3. Bioinformatics and Statistical Analyses

DADA2 analysis was conducted in R [33,34], and the parameters were set as follows: maxEE = c (2,2), minLen = 200, truncLen = c (220,220), minBoot = 80, and Min overlap = 12 bases. To maintain the same sequencing depths, we subsampled the number of reads per sample to the lowest number of reads per samples (48,286). In this study, amplicon sequence variants (ASVs) supported by at least two reads were retained for further analysis. Annotation was done in two steps. First, the initial taxonomical assignment was performed according to the Protist Ribosomal Reference (PR2) database [35]. We assigned taxonomy up to the genus level with the *assignTaxonomy* function at an 80% bootstrap confidence threshold. Second, further taxonomic assignment was done for all ASVs that were not annotated to any species at the first step through using BLAST against the NCBI NT database. Each ASV was annotated to the species in the reference database with the highest percentage identity (PID) at a threshold of 99%. For the species level annotation, all ASVs that were annotated to multiple species were removed.

To study the spatial and temporal distribution of *Margalefidinium* species in Jiaozhou Bay, we extracted 28 ASVs annotated as *Margalefidinium*. After subsampling, each sample contained 48,286 reads. The relative abundance of each ASV was represented by the number of reads in each sample. Phylogenetic trees of *Margalefidinium* ASVs were generated with MEGAX [36] using the maximum likelihood (ML) method and Kimura 2-parameter model with 1000 bootstrap. Surfer 16 (Golden Software LLC, Golden, CO, USA) was used to illustrate sampling sites and spatial distribution of ASV_38. The TCS network of *Margalefidinium* ASVs was constructed by PopART v1.7 [37,38]. Correlation analysis between ASVs and environmental factors was carried out using spearman correlation in the R package corrplot [39]. Bubble charts were drawn with the R package ggplot2 [40]. Line plots were drawn using GraphPad Prism v8 (GraphPad Software, San Diego, CA, USA).

## 3. Results

### 3.1. Biological Diversity of Margalefidinium Species in Jiaozhou Bay

A total of 45,932 ASVs were obtained from the seawater samples collected from the 12 sampling sites in Jiaozhou Bay (Figure 1) in 12 months in 2019, 30,097 ASVs of which were supported by at least two reads [33]. Of these 30,097 ASVs, 28 ASVs were annotated as *Margalefidinium* species, including 1 ASV corresponding to *M. polykrikoides*, 18 ASVs corresponding to *M. fulvescens*, and 9 ASVs annotated to an unknown species of the genus *Margalefidinium* (Table 1). Eighteen distinct ASVs were annotated as *M.*
*fulvescens*, suggesting that this species had high intra-species genetic diversity. Among these ASVs annotated as *M. fulvescens*, ASV_38 showed the highest abundance. Both *M. polykrikoides* and *M. fulvescens* were both annotated as HAB species [5,16] and both have been reported previously in Jiaozhou Bay [5,28].

### 3.2. Spatial-Temporal Dynamics of Margalefidinium Species in Jiaozhou Bay

We found that different *Margalefidinium* species have quite different distribution patterns in Jiaozhou Bay. Furthermore, different ASVs of the same species also displayed different spatial-temporal dynamics. The relative abundance revealed that *Margalefidinium* species have obvious seasonal distribution characteristics. They were mostly detected in early autumn, especially in September, suggesting that *Margalefidinium* species preferred a relatively higher temperature (Figure 2). In terms of geographical distribution, it showed little preference among the 12 sampling sites (Figure 3). In addition, the spatial-temporal dynamics of *Margalefidinium* species were consistent in two different seawater samples in 10–200 μm and in 0.2–10 μm.

For further analysis, ASV_38, with the highest abundance among *Margalefidinium* species, was selected to study the regularities of distribution. It was annotated as *M.*
*fulvescens* and was detected in the seawater samples from July to December (Figure 4a). From July to August, ASV_38 was found only outside the bay, and the number was increasing. By September, it had the largest quantity and distributed in all 12 sampling sites in Jiaozhou Bay. From October to December, the relative abundance of ASV_38 was decreasing, but it was found both inside and outside the bay.

We further analyzed the vertical distribution profiles of ASV_38 (*M. fulvescens*) at different seawater depths in September, considering the strong vertical migration capability of dinoflagellates [41]. In general, the ASV_38 values were highest or close to the highest at the surface (Figure 4), and the minimum values were found at the bottom with a few exceptions (Figure 4b). At sites C3 and D7, wherein both had similar seawater depths (about 15 m), ASV_38 reached maximum values at the surface. However, the minimum values of these two samples sites were different. While the minimum value was found at 10 m water under the surface at the C3 site, the minimum value was found at the bottom at the D7 site. At the D6 site, which was located outside of Jiaozhou Bay, the seawater depth reached 25 m. It also had the maximum of ASV_38 reads at the surface of seawater and the minimum on the bottom, similar to site D7. However, the difference between the bottom and middle layers was substantially reduced compared to the D7 site. The D5 site had the deepest depth among all 12 sampling sites in this expedition in Jiaozhou Bay (≈30 m) and had the unique distribution pattern of ASV_38. From the surface to the bottom, the number of reads decreased first, then increased to the maximum, and finally decreased to the minimum.

### 3.3. Substantial Genetic Diversity of Margalefidinium Species in Jiaozhou Bay

The phylogenetic tree indicated that the 28 ASVs uncovered in the DADA2 analysis formed two major groups (Figure 5a). While most ASVs were annotated to be *M. polykrikoides* or *M. fulvescens*, some were annotated as *Margalefidinium* sp. These ASVs were substantially different from reference 18S rDNA V4 region sequences of these two *Margalefidinium* species. Nevertheless, these ASVs annotated as *Margalefidinium* sp. were clustered closely with *M. polykrikoides* or *M.*
*fulvescens*. The ASV_11381 and ASV_12769 might have a closer relationship with M. *fulvescens*. The ASV_6970 and ASV_1311 seemed to have a closer relationship with *M. polykrikoides*. The phylogenetic network displayed the same results in a different format, highlighting relative abundance of each ASV (Figure 5b). ASV_38 (*M.*
*fulvescens*) was the most abundant haplotype, with most other haplotypes surrounding it, suggesting that many strains with highly similar haplotypes. The ASV_1149, representing *M. polykrikoides*, together with ASV_1311 and ASV_6970, formed the other group. The ASV_6970, which played an important role in connecting the two groups, revealed that these two groups had many sites that are different from each other.

### 3.4. Correlation of Margalefidinium Species with Environmental Factors

To explore the impact of environmental factors on the temporal time course of *Margalefidinium* species, we calculated correlation coefficients between relative abundance of ASVs and environmental variables in Jiaozhou Bay (Figure 6a). It was found that *Margalefidinium* species were positively correlated with temperature. Some of these ASVs were significantly positively correlated with temperature, indicating that temperature was the most important influencing factor for *Margalefidinium* species (Figure 6b). All ASVs showed negative correlations with SiO_3_^2−^, and, in particular, ASV_38 (*M. fulvescens*) was significantly negatively correlated with SiO_3_^2−^ (Figure 6a). Furthermore, all ASVs showed positive correlations with NH_4_^+^, and ASV_38 (*M. fulvescens*) showed significant positive correlation with NH_4_^+^ (Figure 6c). *Margalefidinium* species did not have a significant correlation with other environmental factors, suggesting that *Margalefidinium* species may have strong ecological adaptability and a wide distribution, consistent with previous studies [16,42].

## 4. Discussion

High-throughput metabarcoding is now routine for studying protist diversity and distributions, having been applied in multitude environments at different sampling scales [43], for example, Tara Oceans Expedition [44] and the Ocean Sampling Day [45]. This method provides new opportunities in studying protist diversity, especially for these rare, small, or cryptic species [46]. For time series samples, metabarcoding analysis has also been ongoing for marine investigation, such as in the Mediterranean coast [47]. Thus, metabarcoding analysis could be used for routinely monitoring the HAB species.

Despite extensive research on *Margalefidinium* species, many of which are cosmopolitan HAB species with negative impact on economy and ecosystems, the biodiversity and spatial-temporal dynamics of these species remain poorly understood. Through carrying out metabarcoding analysis of samples collected in Jiaozhou Bay (China) for one entire year, we gained quantitative insight into the spatial-temporal dynamics of *Margalefidinium* species.

### 4.1. Low Biodiversity of Margalefidinium Species but High Intra-Species Genetic Diversity Was Identified in Jiaozhou Bay

According to current taxonomy system, the genus *Margalefidinium* is relatively small, only containing five species [1]. Two species, *M. polykrikoides* and *M. fulvescens*, have been detected in coastal regions in China [23]. In our project, we successfully identified these two species among the 28 ASVs corresponding to species in the genus *Margalefidinium* in Jiaozhou Bay. According to phylogenetic analysis based on 18S rDNA V4 region (Figure 5), these two species formed two sister groups, which was confirmed by previous studies that constructed a phylogenetic tree based on 28S rDNA [17]. Moreover, there were nine ASVs annotated to the genus level; however, they were clustered together with *M. polykrikoides* and *M. fulvescens* instead of forming a separate clade due to larger than 1% nucleotide differences compared with reference sequences in NCBI (Table 1, Figure 5). The annotation result, as well as the phylogenetic analysis, indicated the low biodiversity of *Margalefidinium* species in Jiaozhou Bay.

Interestingly, high intra-species genetic diversity was found in *Margalefidinium* species. Previous studies have proven that the different geographic populations of *M. polykrikoides* around the world are in fact genetically different according to the 28S rDNA gene, and thus they were divided into four ribotypes: the East Asian ribotype, the American/Malaysian ribotype, the Philippines ribotype, and the Mediterranean ribotype [17,28]. In this study, only one ASV (ASV_1149) was specifically annotated as *M. polykrikoides* on the basis of 18S rDNA V4, and two suspected ASVs clustered with *M. polykrikoides* (ASV_1311 and ASV_6970) were annotated as *Margalefidinium* sp. A similar discovery was made in the American/Malaysian ribotype in that despite the genetic similarity of the 28S rDNA gene of clones within the ribotype, there may be ecological differences among these populations [7]. We found 25 ASVs clustered with *M. fulvescens* according to 18S rDNA V4 (Figure 5), and among them, 18 ASVs were annotated as *M. fulvescens* in Jiaozhou Bay. Thus, it was hypothesized that *M. fulvescens* could also be divided into multiple ribotypes, which still need to be confirmed [7]. However, it could not be ruled out that diversities among these 25 ASVs might be due to high copy numbers of ribosomal gene clusters within individual cells of *M. fulvescens*, which was estimated to up to 5620 copies per cell by using the correlation between cell length and rDNA copy number [48]. The study based on 28S rDNA analysis inferred that the variability within individual cells (i.e., variability among polymorphic copies of 28S rDNA) caused both the intra-individual and intra-populational genetic diversities rather than the co-existence of populations of different geographic origins.

### 4.2. Margalefidinium Species in Jiaozhou Bay Demonstrated Strong Temporal Dynamics with a Sharp Peak of Abundance in Early Autumn

In our research of Jiaozhou Bay in 2019, the relative abundance of *Margalefidinium* species demonstrated strong temporal dynamics with a sharp peak of abundance in early autumn (Figure 2). In contrast, the relative abundance of *Margalefidinium* species showed no obvious spatial preference (Figure 3). These results were congruent with a previous study from August to October in 2015 in Jiaozhou Bay, reaching 68,000 cells/L for the total density of *M. polykrikoides* and *M. fulvescens* in the early autumn [5]. Further study found that *Margalefidinium* species showed the highest correlation with water temperature among different environmental factors in this study (Figure 6). The species *M. polykrikoides* have been proven to be eurythermal, well adapted to warm (>20 °C) offshore, possibly tropical or subtropical waters [49,50]. The optimum temperature for growth is relatively high, ranging from 23 to 27 °C, although it can remain viable down to temperatures of approximately 12.5 °C [51]. Distinct from *M. polykrikoides*, *M. fulvescens* appears to be adapted to relatively lower temperatures, associated with cooler coastal upwelling conditions according to the extremely limited data available [7,13]. In Jiaozhou Bay in 2019, the water temperature from January to May ranged between 3 and 17 °C, being neither friendly to the growth of *M. polykrikoides* nor *M. fulvescens*, and therefore we failed to detect them. In summer to early autumn (from June to September), the water temperature rose, and the maximum reached above 26 °C, being consistent with high abundance of two *Margalefidinium* species, especially *M. fulvescens* (ASV_38). The relatively higher abundance of *M. fulvescens* compared to *M. polykrikoides* was consistent with the previous observation, in which *M. fulvescens* appeared to be adapted to lower temperature than *M. polykrikoides* according to the current study [7]. On the other hand, the copy number differences of rDNA per individual cells might be another reason. Our results confirmed that water temperature is a crucial factor for the abundance of *Margalefidinium* species.

Both *M. polykrikoides* and *M. fulvescens* are euryhaline, well adapted to moderate (30–33) salinities sea waters [49,50]. In Jiaozhou Bay in 2019, there was little change in salinity among the months (31.08–32.03), and thus it may not be an explanation for the sharp peak of abundance of *Margalefidinium* species in early autumn.

*Margalefidinium* species exhibits wide flexibility in its nutrient acquisition strategies [7]. Previous studies found that they have a preference for NH_4_^+^ versus NO_2_^−^ and NO_3_^−^ [7], and in our research, we obtained similar results. All ASVs showed positive correlations with NH_4_^+^ concentration in Jiaozhou Bay, and ASV_38 (*M. fulvescens*) showed a significantly positive correlation. In August 2019, before the appearance of the peak abundance of *Margalefidinium* species, the concentration of NH_4_^+^ maximized, which may significantly contribute to the growth of *Margalefidinium* species. Then, in September, when abundance of *Margalefidinium* species went to the maximum, NH_4_^+^ was obviously consumed, lower than the neighbor months. As a result, we inferred that the concentration of NH_4_^+^ is a possible factor for the sharp peak of abundance of *Margalefidinium* species in early autumn. Furthermore, it was supported by previous reviews that both *M. polykrikoides* and *M. fulvescens* can readily utilize organic nitrogen and phosphorus compounds [7], which may explain the result found in our study that *Margalefidinium* species do not have significant correlation with some of the environmental factors, such as NO_2_^−^ and NO_3_^−^. Thus, we next need to pay more attention to the organic substance carried into Jiaozhou Bay instead of the inorganic nutrient to better explain the sharp peak of abundance of *Margalefidinium* species in early autumn.

## 5. Conclusions

Our data provided the first analysis of monthly and spatial dynamics of *Margalefidinium* species in Jiaozhou Bay using the metabarcoding approach. In this study, we detected two *Margalefidinium* species (*M. polykrikoides* and *M. fulvescens*) with high genetic diversity in Jiaozhou Bay. *M. polykrikoides* and *M. fulvescens* showed a strong temporal shift with a sharp peak of abundance in early autumn around the whole year, while they did not show significant preference among different sites in Jiaozhou Bay. Our results also suggested the temperature might be the main contributor for their obvious temporal dynamics. Thus, the summer and early autumn might be the possible outbreak time of *Margalefidinium* species blooms in Jiaozhou Bay, which requires more attention.

## Figures and Tables

**Figure 1 ijerph-18-11637-f001:**
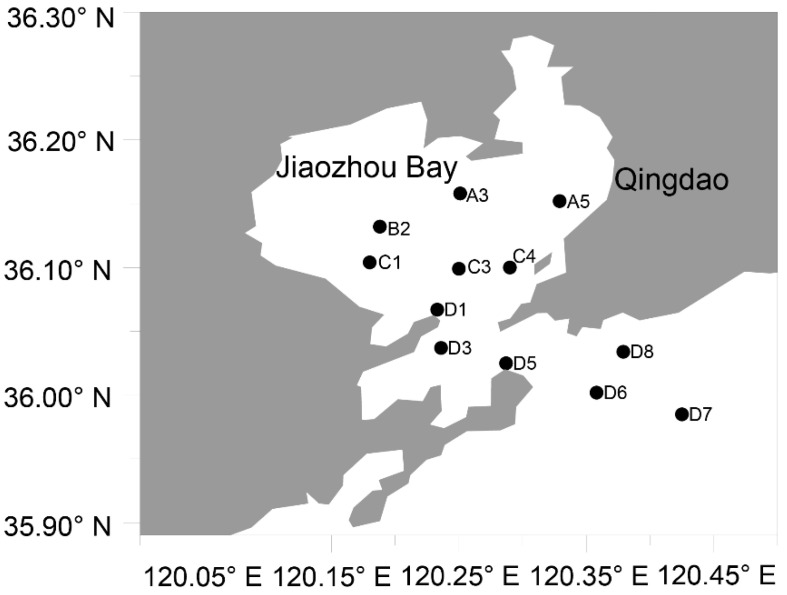
Sampling sites in Jiaozhou Bay.

**Figure 2 ijerph-18-11637-f002:**
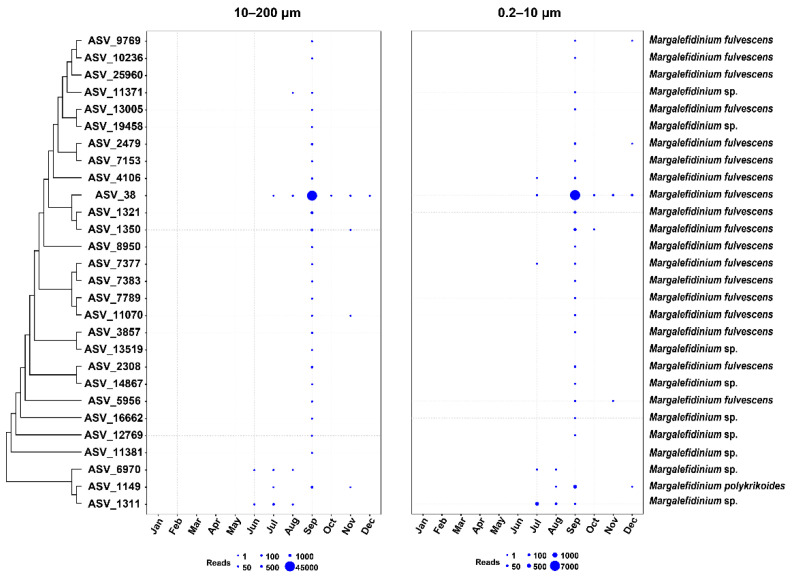
Temporal dynamics of *Margalefidinium* species for the surface water in Jiaozhou Bay.

**Figure 3 ijerph-18-11637-f003:**
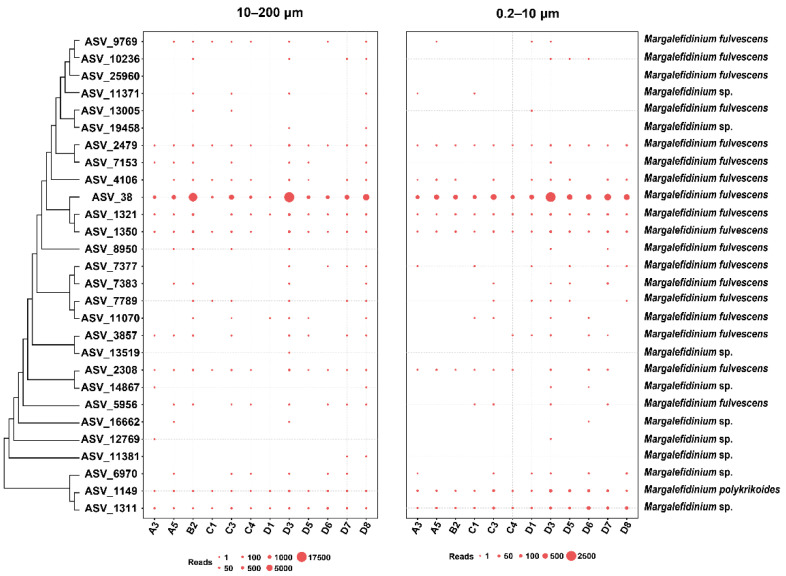
Spatial variations of *Margalefidinium* species for the surface water in Jiaozhou Bay.

**Figure 4 ijerph-18-11637-f004:**
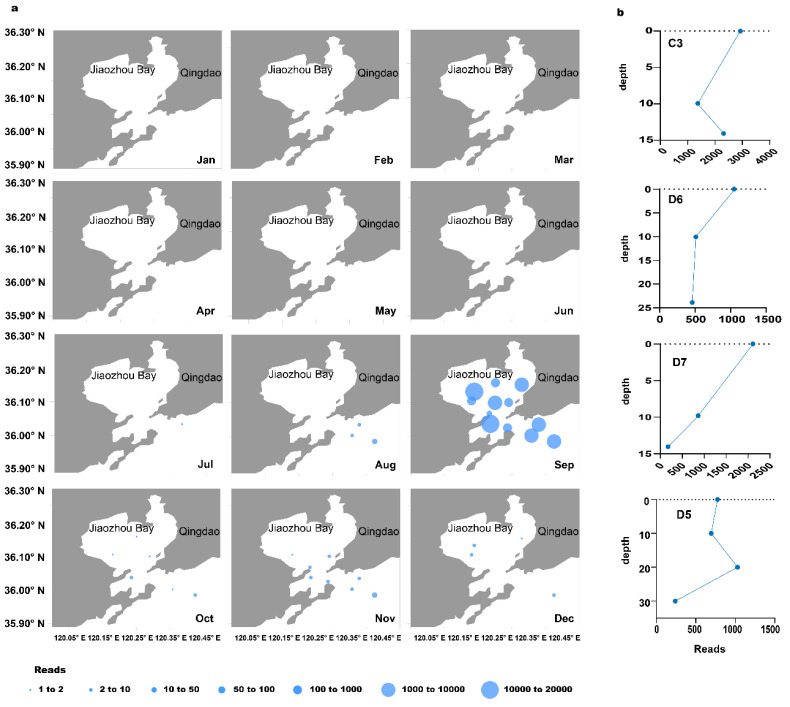
The spatial and temporal distributions of *M. fulvescens* (ASV_38). (**a**) Horizontal distributions of *M. fulvescens* for 12 months in 10–200 μm fraction of surface water in Jiaozhou Bay. (**b**) The vertical distribution of *M. fulvescens* in September for C3, D6, D7, and D5 sites.

**Figure 5 ijerph-18-11637-f005:**
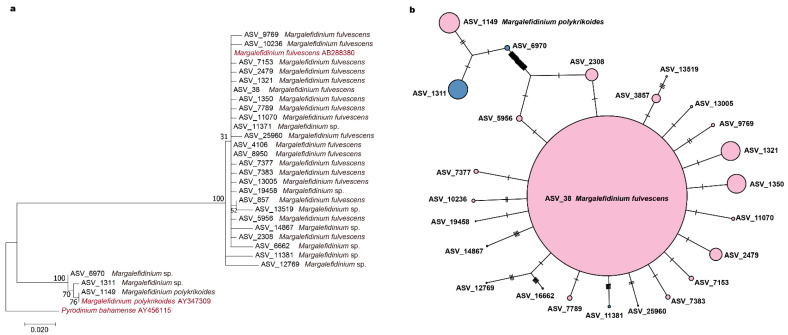
The phylogenetic tree (**a**) and TCS network of *Margalefidinium* species in Jiaozhou Bay (**b**). The pink points refer to the ASVs annotated as being *M. fulvescens* or *M. polykrikoides* and the blue points refer to the ASVs annotated as being from the *Margalefidinium* genus. The size of dots is in correspondence to the relative abundance of *Margalefidinium* ASV in Jiaozhou Bay.

**Figure 6 ijerph-18-11637-f006:**
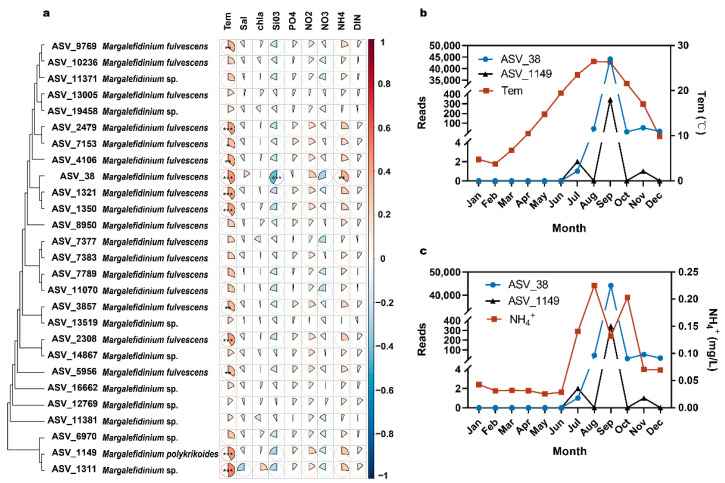
(**a**) Correlation of *Margalefidinium* species with environmental factors in 10–200 μm fraction of the surface water in Jiaozhou Bay. The pies with * indicate that *p*-value < 0.05, the pies with ** indicate *p*-value < 0.01, the pies with *** indicate *p*-value < 0.001. (**b**) Temporal variations of ASV_38 (*M. fulvescens*) abundances and ASV_1149 (*M. polykrikoides*) abundances, and temperature for the 10–200 μm fraction for the surface water in Jiaozhou Bay. (**c**) Temporal variations of ASV_38 (*M. fulvescens*) abundances and ASV_1149 (*M. polykridoides*) abundances, and NH_4_^+^ concentrations for the 10–200 μm fraction for the surface water in Jiaozhou Bay.

**Table 1 ijerph-18-11637-t001:** List of ASVs annotated as *Margalefidinium* species detected in this study in Jiaozhou Bay.

ASV_Name	Species	Group
ASV_38	*Margalefidinium fulvescens*, AB288380,100	mvs1
ASV_1321	*Margalefidinium fulvescens*, AB288380,99.732	mvs1
ASV_1350	*Margalefidinium fulvescens*, AB288380,99.464	mvs1
ASV_2308	*Margalefidinium fulvescens*, AB288380,99.732	mvs1
ASV_2479	*Margalefidinium fulvescens*, AB288380,99.732	mvs1
ASV_3857	*Margalefidinium fulvescens*, AB288380,99.732	mvs1
ASV_4106	*Margalefidinium fulvescens*, AB288380,99.732	mvs1
ASV_5956	*Margalefidinium fulvescens*, AB288380,99.464	mvs1
ASV_7153	*Margalefidinium fulvescens*, AB288380,99.732	mvs1
ASV_7377	*Margalefidinium fulvescens*, AB288380,99.732	mvs1
ASV_7383	*Margalefidinium**fulvescens*, AB288380,99.732	mvs1
ASV_7789	*Margalefidinium fulvescens*, AB288380,99.732	mvs1
ASV_8950	*Margalefidinium fulvescens*, AB288380,99.732	mvs1
ASV_9769	*Margalefidinium fulvescens*, AB288380,99.464	mvs1
ASV_10236	*Margalefidinium fulvescens*, AB288380,99.464	mvs1
ASV_11070	*Margalefidinium fulvescens*, AB288380,99.732	mvs1
ASV_13005	*Margalefidinium fulvescens*, AB288380,99.732	mvs1
ASV_25960	*Margalefidinium fulvescens*, AB288380,99.464	mvs1
ASV_1149	*Margalefidinium**polykrikoides*, AY347309,100	1vs1
ASV_1311	*Margalefidinium* sp.	genus level
ASV_6970	*Margalefidinium* sp.	genus level
ASV_11371	*Margalefidinium* sp.	genus level
ASV_11381	*Margalefidinium* sp.	genus level
ASV_12769	*Margalefidinium* sp.	genus level
ASV_13519	*Margalefidinium* sp.	genus level
ASV_14867	*Margalefidinium* sp.	genus level
ASV_16662	*Margalefidinium* sp.	genus level
ASV_19458	*Margalefidinium* sp.	genus level

## Data Availability

The sequencing results (raw data) have been submitted to NCBI, and the BioProject numbers are PRJNA577777 and PRJNA733859.

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
