# Peer review of "Biodiversity and Spatial-Temporal Dynamics of Margalefidinium Species in Jiaozhou Bay, China"

_ijerph, 2021, doi:10.3390/ijerph182111637_

Round 1

Reviewer 1 Report

The manuscript evaluates the occurrence of two relevant species of harmful dinoflagellates in Jiaozhou Bay during the year 2019. The authors show interesting results using metabarcoding applying it to two size fractions 0.2 µm-10 µm and 10 µm-200 µm. Some authors are currently defending the use of metabarcoding for routine monitoring of harmful algal blooms. It will be very useful to include a paragraph in the discussion related to the applicability of the method for routine monitoring taking into account the need to obtain result between 1 to two days after sampling when samples are in the framework of a routine HAB monitoring. The manuscript refer to the great potential of metabarcoding for phytoplankton monitoring in line 95.
The line number is included in the text to facilitate finding the paragraph.
1)    Jiaozhou Bay is an ideal area for HAB researches, and was the earliest marine survey area in China. 21
Do you mean first marine research station in China?

2)    Two ichthyotoxic Margalefidinium 24 species i.e. M. polykrikoides and M. fulvescens were identified with potentially high genetic diversity 25
Replace for:
Two harmful Margalefidinium 24 species i.e. M. polykrikoides and M. fulvescens were identified with potentially high genetic diversity 25

3)    M. catenatum, M. citron and M. flavum, was recently isolated from the genus Cochlodinium 38
Replace for:
M. catenatum, M. citron and M. flavum, were recently transferred from the genus from the genus Cochlodinium 38

4)    expanding in scope, respectively [7]. The first blooms of M. polykrikoides were found in 53
Replace for:
expanding in scope, respectively [7]. The first blooms of M. polykrikoides were detected in 53

5)    Although mechanisms of M. polykrikoides on killing fish has 64 yet to been fully validated, they may involve many substances, such as reactive oxygen 65 species, hemolytic and neurotoxic-like substances, and extracellular mucoid polysaccha-
Replace for: 
Although the mechanisms of toxicity of M. polykrikoides to fish have 64 yet to been fully

6)    M. fulvescens was 69 difficult to culture and had relatively little studies on its harmful effects, some studies 70 suggested that M. fulvescens possesses ichthyotoxic properties similar to those of M. 71 polykrikoides [4,16]. 72
Replace for
M. fulvescens is 69 difficult to culture and there are relatively few studies on its harmful effects, some studies 70 suggested that M. fulvescens possesses ichthyotoxic properties similar to those of M. 71 polykrikoides [4,16]. 72

7)    Resting cysts have been considered a fundamental attrib- 76ute of dinoflagellate life cycles, that in M. polykrikoides is also a research hotspot [18].
Authors should provide a more appropriate reference for the species or place the citation number 18 after “cycles”. It is also preferable to use “a research ‘hot’ topic” instead of research hotspot since the term is often used in the sense of a group of researchers 

8)    quickly added after taking out of the liquid nitrogen It was then cut into pieces using 154
full stop is missing before It

9)    Line 213 Table 1 The information that appears in the 3 last columns is always the same, the table could be simplified.

10)    The relative abundance revealed that Margalefi- 217
Authors must include in the methodology a paragraph describing how they calculated the relative abundance

11)    The figures are too small and too difficult to read

12)    The within individual cells of M. fulvescens, which was estimated to up to 5620 copies/cell by 333 using the correlation between cell length and rDNA copy number [28,45]. 

In reference 28 the number of copies/cell is lower than 5620, 5620 is only in reference 45. Authors should modify the phrase to cite properly both references.

Author Response

The manuscript evaluates the occurrence of two relevant species of harmful dinoflagellates in Jiaozhou Bay during the year 2019. The authors show interesting results using metabarcoding applying it to two size fractions 0.2 µm-10 µm and 10 µm-200 µm. Some authors are currently defending the use of metabarcoding for routine monitoring of harmful algal blooms. It will be very useful to include a paragraph in the discussion related to the applicability of the method for routine monitoring taking into account the need to obtain result between 1 to two days after sampling when samples are in the framework of a routine HAB monitoring. The manuscript refer to the great potential of metabarcoding for phytoplankton monitoring in line 95.

Response: Thank you very much for your advice, and we have added the application of metabarcoding analysis in the discussion. Metabarcoding analysis might be not suitable for HAB monitoring between 1 to two days after sampling due to the long time used for sequencing, which is currently done by sequencing company.

The line number is included in the text to facilitate finding the paragraph.
1)    Jiaozhou Bay is an ideal area for HAB researches, and was the earliest marine survey area in China. 21
Do you mean first marine research station in China?

Response: This statement was based on information from the review of “Jiaozhou Bay is the earliest marine survey area in China” from the review (Song et al., 2020). To avoid the possible confusion, we have modified the expression in the manuscript.

Reference: Song, J; Yuan, H; Li, X; Duan, L; Ecological environment evolution and nutrient variations in Jiaozhou Bay (Chinese with English abstract). Marine Sciences 2020, 44,8, doi: 10.11759/hykx20200220001.

2)    Two ichthyotoxic Margalefidinium 24 species i.e. M. polykrikoides and M. fulvescens were identified with potentially high genetic diversity 25
Replace for:
Two harmful Margalefidinium 24 species i.e. M. polykrikoides and M. fulvescens were identified with potentially high genetic diversity 25

Response: Corrected.

3)    M. catenatum, M. citron and M. flavum, was recently isolated from the genus Cochlodinium 38
Replace for:
M. catenatum, M. citron and M. flavum, were recently transferred from the genus Cochlodinium 38

Response: Corrected.

4)    expanding in scope, respectively [7]. The first blooms of M. polykrikoides were found in 53
Replace for:
expanding in scope, respectively [7]. The first blooms of M. polykrikoides were detected in 53

Response: Corrected.

5)    Although mechanisms of M. polykrikoides on killing fish has 64 yet to been fully validated, they may involve many substances, such as reactive oxygen 65 species, hemolytic and neurotoxic-like substances, and extracellular mucoid polysaccha-
Replace for: 
Although the mechanisms of toxicity of M. polykrikoides to fish have 64 yet to been fully

Response: Corrected.

6)    M. fulvescens was 69 difficult to culture and had relatively little studies on its harmful effects, some studies 70 suggested that M. fulvescens possesses ichthyotoxic properties similar to those of M. 71 polykrikoides [4,16]. 72
Replace for
M. fulvescens is 69 difficult to culture and there are relatively few studies on its harmful effects, some studies 70 suggested that M. fulvescens possesses ichthyotoxic properties similar to those of M. 71 polykrikoides [4,16]. 72

Response: Corrected.

7)    Resting cysts have been considered a fundamental attrib- 76ute of dinoflagellate life cycles, that in M. polykrikoides is also a research hotspot [18].
Authors should provide a more appropriate reference for the species or place the citation number 18 after “cycles”. It is also preferable to use “a research ‘hot’ topic” instead of research hotspot since the term is often used in the sense of a group of researchers 

Response: Thank you for your advice, and we have modified the expression.

8)    quickly added after taking out of the liquid nitrogen It was then cut into pieces using 154
full stop is missing before It

Response: Corrected.

9)    Line 213 Table 1 The information that appears in the 3 last columns is always the same, the table could be simplified.

Response: We have deleted the three last columns due to their information have been described in the manuscript.

10)    The relative abundance revealed that Margalefi- 217
Authors must include in the methodology a paragraph describing how they calculated the relative abundance

Response: We have added the description of “relative abundance of Margalefidinium ASVs in the part of “2.3. Bioinformatics and Statistical analyses” (line 192-194).

11)    The figures are too small and too difficult to read.

Response: Thank you for your kind advice, we have modified the pictures.

12)    The within individual cells of M. fulvescens, which was estimated to up to 5620 copies/cell by 333 using the correlation between cell length and rDNA copy number [28,45]. 

In reference 28 the number of copies/cell is lower than 5620, 5620 is only in reference 45. Authors should modify the phrase to cite properly both references.

Response: Thank you very much, we have modified the reference in the manuscript.

Reviewer 2 Report

The manuscript by Liu et al. refers to a metabarcoding analysis to uncover Margalefidinium species spatial and temporal dynamics. Though the work presented is valid and well-presented it lacks some clarifications. For one why the need to evaluate the spatial and temporal dynamic of this HAB’s species? I mean is there any data namely microscopy observation and counts on this Bay or elsewhere? And why a 12 month period sampling encompassing a winter and spring? Has a bloom-forming species when does it forms blooms in this region? Where there any blooms formed during the sampling period? What about toxicity is there any reports on toxicity by this species in Jiaozhou Bay? Why were no toxins measured in this study? I think the authors need to address these issues and discuss them with their data.

Another remark regards M. polykrikoides and M. fulvescens bloom-forming in Jiaozhou Bay. Are they common in this region? The authors should clarify this topic since it is not obvious in my opinion after reading the manuscript.

Another issue does the data collected from the study enables prediction of blooms or seasonality of Margalefidinium species?

Minor comments:

Line 44 – “photoreceoption” or photoreception?

Line 168 – word “by” is repeated.

Author Response

The manuscript by Liu et al. refers to a metabarcoding analysis to uncover Margalefidinium species spatial and temporal dynamics. Though the work presented is valid and well-presented it lacks some clarifications. For one why the need to evaluate the spatial and temporal dynamic of this HAB’s species? I mean is there any data namely microscopy observation and counts on this Bay or elsewhere?

Response: Hu et al., (2020) and Lin et al., (2020) have validated the presence and high abundance of M. polykrikoides and M. fulvescens in the Jiaozhou Bay by microscopy observation.

Reference:

Hu, Z.; Deng, Y.; Li, Y.; Tang, Y.Z. The morphological and phylogenetic characterization for the dinoflagellate Margalefidinium fulvescens (=Cochlodinium fulvescens) isolated from the Jiaozhou Bay, China. Acta Oceanologica Sinica 2018, 37, 11-17, doi:10.1007/s13131-018-1295-0.

Lin, S.; Hu, Z.; Deng, Y.; Shang, L.; Gobler, C.J.; Tang, Y.Z. An assessment on the intrapopulational and intraindividual genetic diversity in LSU rDNA in the harmful algal blooms-forming dinoflagellate Margalefidinium (= Cochlodinium) fulvescens based on clonal cultures and bloom samples from Jiaozhou Bay, China. Harmful Algae 2020, 96, 101821, doi:10.1016/j.hal.2020.101821.

And why a 12 month period sampling encompassing a winter and spring?

Response: We conducted the surveys monthly in 2019, including winter (January, February, and December) and spring (March, April and May).

Has a bloom-forming species when does it forms blooms in this region?

Response: No severe HABs caused by Margalefidinium species in the Jiaozhou bay has been reported. Hu et al., (2020) found that the cell density of M. fulvescens and M. polykrikoides in the Jiaozhou Bay ranged from 2,000 to 6,8000 cells/L, which might be a slight bloom without obvious deleterious effects from August to October in 2015.

Reference:

Hu, Z.; Deng, Y.; Li, Y.; Tang, Y.Z. The morphological and phylogenetic characterization for the dinoflagellate Margalefidinium fulvescens (=Cochlodinium fulvescens) isolated from the Jiaozhou Bay, China. Acta Oceanologica Sinica 2018, 37, 11-17, doi:10.1007/s13131-018-1295-0.

Where there any blooms formed during the sampling period?

Response: We didn’t find the blooms during the sampling period.

What about toxicity is there any reports on toxicity by this species in Jiaozhou Bay?

Response: Wang et al., (2020) studied the toxins of M. fulvescens, and showed that M. polykrikoides of the Jiaozhou Bay strains (MPJZB-C3 and MPJZB-D6) showed lower toxicity than that of USA strain (CP1) and Malaysia strain (MPCoKK23).

Reference:

Wang, H.; Hu, Z.; Shang, L.; Leaw, C.P.; Lim, P.T.; Tang, Y.Z. Toxicity comparison among four strains of Margalefidinium polykrikoides from China, Malaysia, and USA (belonging to two ribotypes) and possible implications. Journal of Experimental Marine Biology and Ecology 2020, 524, 151293, doi:10.1016/j.jembe.2019.151293.

Why were no toxins measured in this study? I think the authors need to address these issues and discuss them with their data.

Response: This study focused on the biodiversity and spatial-temporal dynamics of Margalefidinium species in the Jiaozhou Bay, thus the toxins measurement were not conducted in this project.

Another remark regards M. polykrikoides and M. fulvescens bloom-forming in Jiaozhou Bay. Are they common in this region? The authors should clarify this topic since it is not obvious in my opinion after reading the manuscript.

Response: The topic of this manuscript was to study the biodiversity and spatial-temporal dynamics of Margalefidinium species in the Jiaozhou Bay using the metabarcoding analysis. There was no detailed investigation about spatial-temporal variations of Margalefidinium species in the Jiaozhou Bay monthly for the whole year previously. Through this study, we found that M. polykrikoides and M. fulvescens were common in this region during autumn to early autumn.

Another issue does the data collected from the study enables prediction of blooms or seasonality of Margalefidinium species?

Response: Margalefidinium species in the Jiaozhou Bay demonstrated strong temporal dynamics with a sharp peak of abundance in early autumn. Thus, early autumn may be the possible outbreak time of Margalefidinium species blooms in the Jiaozhou Bay, and we have added this information in the manuscript.

Minor comments:

Line 44 – “photoreceoption” or photoreception?

Line 168 – word “by” is repeated.

Response: Thank you for your advice, we have corrected these expressions.

Round 2

Reviewer 2 Report

The authors have improved the manuscript and now its suitable for publication.

Author Response

Thank you very much for your kind comments.